# Development and Evaluation of Molecular Pen-Side Assays without Prior RNA Extraction for Peste des Petits Ruminants (PPR) and Foot and Mouth Disease (FMD)

**DOI:** 10.3390/v14040835

**Published:** 2022-04-17

**Authors:** David Edge, Mana Mahapatra, Shona Strachan, James Turton, Ryan Waters, Camilla Benfield, Nathan Nazareth, Felix Njeumi, Nelson Nazareth, Satya Parida

**Affiliations:** 1BioGene Limited, 6 The Business Centre, Harvard Way, Kimbolton PE28 0NJ, UK; david@biogene.com (D.E.); j.turton@bgresearchltd.com (J.T.); nathan@biogene.com (N.N.); nelson@Biogene.com (N.N.); 2The Pirbright Institute, Ash Road, Pirbright, Woking, Surrey GU24 ONF, UK; manamahapatra1964@gmail.com (M.M.); shona.strachan@gmail.com (S.S.); ryan.waters@pirbright.ac.uk (R.W.); 3Royal Veterinary College, University of London, Hawkshead Lane, North Mimms, Hatfield AL9 7TA, UK; camilla.benfield@fao.org; 4Food and Agriculture Organization of the United Nations (FAO), Viale delle Terme di Caracalla, 00153 Rome, Italy; felix.njeumi@fao.org

**Keywords:** pen-side PCR, peste des petits ruminants, foot and mouth disease, PPR, FMD, PCR, RNA extraction

## Abstract

Animal diseases such as peste des petits ruminants (PPR) and foot and mouth disease (FMD) cause significant economic losses in endemic countries and fast, accurate in-field diagnostics would assist with surveillance and outbreak control. The detection of these pathogens is usually performed at reference laboratories, tested using assays that are recommended by The World Organisation for Animal Health (OIE), leading to delays in pathogen detection. This study seeks to demonstrate a proof-of-concept approach for a molecular diagnostic assay that is compatible with material direct from nasal swab sampling, without the need for a prior nucleic acid extraction step, that could potentially be applied at pen-side for both PPR and FMD. The use of such a rapid, low-cost assay without the need for a cold chain could permit testing capacity to be established in remote, resource limited areas and support the surveillance activities necessary to meet the goal of eradication of PPR by 2030. Two individual assays were developed that detect > 99% of PPR and FMD sequences available in GenBank, demonstrating pan-serotype FMD and pan-lineage PPR assays. The ability for the BioGene XF reagent that was used in this study to lyse FMD and PPR viruses and amplify their nucleic acids in the presence of unprocessed nasal swab eluate was evaluated. The reagent was shown to be capable of detecting the viral RNA present in nasal swabs collected from naïve and infected target animals. A study was performed comparing the relative specificity and sensitivity of the new assays to the reference assays. The study used nasal swabs collected from animals before and after infection (12 cattle infected with FMDV and 5 goats infected with PPRV) and both PPR and FMD viral RNA were successfully detected two to four days post-infection in all animals using either the XF or reference assay reagents. These data suggest that the assays are at least as sensitive as the reference assays and support the need for further studies in a field setting.

## 1. Introduction

Animal diseases such as peste des petits ruminants (PPR) and foot and mouth disease (FMD) pose a risk to both small- and large-scale farmers across the globe. Economic losses through animal mortality and reductions in meat and milk production amount to $1.4–2.1 billion (USD) for PPR and over $6 billion (USD) for FMD [1,2,3]. With a global population set to reach 8 billion by 2025, the need for increased food security is ever present.

PPR virus (PPRV), the etiological agent of PPR, is a negative sense single stranded RNA virus from the *Morbillivirus* genus, a member of the *Paramyxoviridae* family. Since its discovery in Côte d’Ivoire in 1942 [4] the virus has spread across Western and Eastern Africa through to the Middle East and into Asia, many countries of which are endemic for the virus [5,6]. PPR was first detected in Europe in Georgia in 2016 [7,8], and further spread into Bulgaria where it was detected in 2018 [9], making the virus a growing threat to European farmers. PPRV most commonly infects sheep and goats and is characterised by pyrexia, ocular and nasal discharges which may become mucopurulent over the course of the infection, diarrhoea, respiratory problems, and lesions in the oral cavity. The disease has a high level of morbidity and mortality and transmission occurs mainly through close contact of infected and naïve animals. There is a single serotype of PPRV, but isolates can be grouped into four genetic lineages corresponding to their geographical location; lineage I and II originating in West Africa, lineage III in East Africa and the Middle East, and lineage IV in Asia [10], although lineage IV is now reported in most African countries and is becoming the dominant strain worldwide. Vaccination allows life-long immunity against all lineages of the virus, which has facilitated a joint initiative between the World Organisation for Animal Health (OIE) and Food and Agriculture Organisation (FAO) to develop a control program to eradicate PPR by 2030 [11].

Foot and mouth disease virus (FMDV), the etiological agent of FMD, is a picornavirus in the genus *Aphthovirus*. The disease is an acute and severely systemic disease affecting cloven-hoofed animals, most notably livestock including cattle, goats, sheep, and pigs, being endemic across much of Africa, the Middle East, and Asia [12]. Acute clinical signs include pyrexia and appearance of vesicles in and around the epithelium of the mouth, muzzle/snout, the teats of the udders, and the interdigital space of the hooves [13]. Clinical signs generally resolve completely in around two to three weeks, however FMDV may be detected in the pharynx of up to 50% of ruminants after resolution of clinical signs termed “carrier animals” which may be an ongoing source of infection to infect naïve populations [14]. Currently, North America, Western Europe, Australia, and New Zealand are all FMD-free regions. FMDV is comprised of seven serotypes; A, O, C, Asia 1, and South African Territories (SAT) 1, SAT2, and SAT3. The SAT serotypes are normally restricted to sub-Saharan Africa, Asia 1 is present in Asia, while serotypes A and O have the widest distribution across most of Africa, the Middle East, and Asia [15]. Serotype C has largely been observed on the Indian sub-continent, but has caused outbreaks in Europe, Africa, and South America and, is now considered extinct [16].

The last large European outbreak of FMD occurred predominantly in the UK in 2001 which resulted in costs of an estimated €6 billion through animal destruction and losses in meat and milk production [2,17]. This outbreak cost some European countries their FMD-free status and reinforced the necessity for accurate diagnostic testing.

A central component of any disease control program, in particular contagious viral diseases, is the availability of sensitive, specific, and rapid diagnostic testing of suspect cases. Currently, pen-side tests and laboratory-based molecular assays are available for PPR and FMD. Pen-side testing includes antigen detection lateral flow devices (LFD) and loop-mediated isothermal amplification (LAMP) which can rapidly test potentially infected animals, but lack high levels of sensitivity and/or specificity and, have not been tested in large scale, neither in the laboratory or in the field [18,19,20,21,22,23,24,25]. Certain molecular assays such as RT-qPCR (reverse transcription quantitative polymerase chain reaction) are currently the OIE “gold standard” for diagnosis of PPR and FMD [26,27]. However, these assays require animal samples to be sent to centralized laboratories, where multistep workflows and processes are required by skilled staff to carry out the testing. Very little work has been published which combines the functionality and rapidity of pen-side testing with the specificity of RT-qPCR. A method which removes the requirement for laboratory processing, particularly the need for nucleic acid extraction, could be used by non-expert users as a pen-side test in remote areas. This is particularly important for disease eradication programmes, such as PPR, when the last foci of infection are likely to occur in hard-to-reach areas. The XF reagent being evaluated in this study is able to increase assay sensitivity by being able to perform multiple cycles of reverse transcription, which is important when the input of unprocessed biological sample is low. The proposed assays can be run off the grid and use lyophilized reagents that only require a resuspension step and the addition of the crude biological samples in a closed tube, allowing the tests to be used in remote areas with minimal training.

The present study describes the design and development of a novel direct RT-qPCR approach to detect both PPRV and FMDV separately, and its ability to detect virus in comparison to established RT-qPCR protocols [26,27].

## 2. Materials and Methods

### 2.1. Cells and Viruses

Four PPR viruses representing the four lineages (I–IV) were selected for use in this study (Table 1). Similarly, six FMD viruses representing the currently circulating six serotypes (Table 1) were used in this study. The sensitivity of the designed PPR and FMD assays was evaluated using cell culture -grown viruses. For each lineage of PPRV (I—Ivory Coast/89, II—Nigeria/75/1, III—Sudan/72, IV—Morocco/2008) the isolates were cultured on VDS (Vero Dog SLAM) cells, while an isolate for each FMDV serotype (A (A22/Iraq/24/64), O (O1/Manisa), Asia1 (Asia1 Shamir), SAT1 (SAT1/TAN/22/2012), SAT2 (SAT2/Eritrea/98) and SAT3 (SAT3/ZIM/83)) was cultured on BHK21 cells (Table 2). Once infected cells showed extensive cytopathic effects (CPE), flasks were frozen and the virus harvested upon thawing. Virus concentrations in TCID_50_/mL in the consequent lysate were measured by titration on respective cell monolayers.

### 2.2. Collection of Swabs from Infected Animals

A total of ten cattle were vaccinated with an inactivated serotype O FMD vaccine as part of a separate vaccine assessment study, with an additional 2 animals remaining unvaccinated. Twenty-one days later, all the animals were challenged intra-dermolingually with a serotype O FMDV virulent isolate by injection of the virus into the tongue epithelium. One day after challenge, all the animals exhibited an elevated rectal temperature and signs of replication at the injection site. Nasal swabs were collected from both naive and vaccinated cattle before and thereafter daily following intradermolingual challenge up to 3 days post challenge (dpc) which was the time at which the control animals showed signs of systemic generalization of vesicles. For the PPRV, nasal swabs were collected from goats experimentally infected with a PPRV isolate from lineage IV (Morocco/2008). The swabs were collected daily/on alternative days from 0 up to 14 days post-infection (dpi) as described previously [28]—dependent on the requirement to cull individual animals on ethical grounds. The swabs were stored at −70 °C until use.

### 2.3. Primer and Assay Design

#### 2.3.1. PPR Assay

The PPRV sequence dataset consisting of both full genome and N-gene open reading frames (ORF) were downloaded from GenBank (accessed on 22 November 2018), representing all four lineages and a wide geographical distribution. From the full genome sequences, the N-gene region was extracted and a total of 300 N-gene sequences were aligned using Geneious V2019.1.1 (Geneious, Auckland, New Zealand). The new assay was to be centered on the location of the primers and probes of the current gold standard reference assay described by Batten and colleagues [26]. The N gene region was modelled for secondary structure under the intended assay conditions using Visual OMP v7.8 software (DNA Software Inc., Ann Arbor, MI, USA) and then the primers and probe were designed manually and subsequently modelled using Visual OMP which uses a multi-state model to accurately predict the binding efficiencies and the potential for any unintended hybridization [29,30]. The 82 bp assay consists of six forward primers and seven reverse primers, covering over 99.9% of available sequences with 100% complementarity, verified by subsequent BLAST analysis of the sequences. During the BLAST analyses, no further potential primer/probe sequences were identified. A single dual-labelled hydrolysis probe covering all the lineages and sequences, and labelled with 5′-CY5 and 3′-BHQ2 was designed. The 82 bp assay is located between bases 395 and 477 in the N-terminus of the N gene sequence (accession no. KY885100) and is shown in Figure 1, with the nine 5′ bases of the primers and probes highlighted. The exact sequences of the primers and probes could not be disclosed because of issues with intellectual property. In silico specificity and sensitivity analyses were performed using Visual OMP and its THERMOBLAST functionality, which identifies candidates for off-target binding by sequence similarity and then attempts to model their hybridization to ensure no unintended hybridizations can occur.

#### 2.3.2. FMD Assay

A total of 1044 full genome sequences were downloaded from GenBank (accessed on 22 November 2018), representing all seven serotypes and a wide geographical distribution. The 3D gene region of the sequences was extracted from the full genome and aligned using Geneious V2019.1.1. The primers and probes of the new assay are located close to that of the current gold standard reference assay described by Callahan and colleagues [27]. The 3D gene region was modelled for secondary structure under the intended assay conditions using Visual OMP and then the primers and probe were designed manually and subsequently modelled using Visual OMP. The 63 bp assay consists of six forward primers and seven reverse primers, covering over 99.9% of the available sequences with 100% complementarity, verified by subsequent BLAST analysis. Two dual-labelled hydrolysis probes covering all FMDV serotypes and labelled with 5′-CY5 and 3′-BHQ2 were designed. The assay is located between bases 6823 and 6884 of GenBank accession AF189157. The location of the 63 bp assay is shown in Figure 2, with the nine 5′ bases of the primers and probes highlighted. In silico specificity and sensitivity analyses were performed similarly as described for the PPR assay.

### 2.4. Assay Optimization

The two new assays were optimized by running 50 µL reactions using the BioGene XF reagent (BioGene Ltd., Kimbolton, UK) and 50 copies of in vitro transcribed (IVT) RNA on an Applied Biosystems Quantstudio 5 instrument (Applied Biosystems, Waltham, MA, USA). The IVT RNA was prepared for genotypes represented by the GenBank accession numbers; KJ8675543, KJ867545, and KX421388 for PPRV and AJ593141, AF308157 and AY593847 for FMDV. The IVT RNA was synthesized by ordering 6 individual IDT gBlocks (IDT, Leuven, Belgium), DNA templates with a T7 promoter sequence upstream of a 200 bp sequence containing the region where the assays were located. RNA was subsequently prepared using the MEGA shortscript kit (Thermofisher, Waltham, USA) following the manufacturer’s instruction. The RNA was then treated with 1 µL of TURBO DNase (Thermofisher, Waltham, USA) for 15 min at 37 °C to degrade any template DNA. The IVT RNA was cleaned up using an NucleoSpin RNA plus kit (Macherey Nagel, Dueren, Germany) following the manufacturer’s instruction. Lastly, the RNA was quantified using Nanodrop ONE (ThermoFisher, Waltham, USA) and stored frozen at high concentration; diluted aliquots were made freshly for all experiments. The variables that were tested were primer mix concentration, probe concentration and temperature gradients for each of the assay steps. The primer concentrations that were used were equimolar forward and reverse and of each individual primer within the mix, a range of 100 nM to 400 nM in 50 nM increments were examined. The probe concentrations were equimolar for the FMDV design for the two probes. The probe ranges that were tested were 60 to 200 nM in 10 nM increments. For the temperature gradient studies, the Veriflex feature of the Quantstudio 5 instrument was used, temperature ranges from 3 degrees to 8 degrees below the predicted melting temperature (Tm) were compared for each of the reverse transcription, cyclical reverse transcription and PCR stages. 

### 2.5. Assay Conditions

#### 2.5.1. PPR Assay 

RT-qPCR was carried out using the appropriate primers and probes and the BioGene XF reagent system, in one of two assay formats. 

(1)Unless stated otherwise, 25 µL reaction volumes were prepared consisting of 1 µL of template, 15.2 µL of nuclease free water, 6.25 µL of 4* XF buffer, 0.875 µL of 10 µM forward primer mix (350 nm of each final conc), 0.875 µL of 10 µM reverse primer mix (350 nM of each final conc), 0.3 µL of 10 µM probe (120 nM final conc), and 0.5 µL XF RTA enzyme. The assay conditions were as follows: 1 cycle of 93 °C for 5 s, 68 °C for 45 s, 72 °C for 150 s, then 7 cycles of 93 °C for 4 s, 68 °C for 30 s and 72 °C for 45 s, a single 93 °C for 10 s, and lastly, 35 cycles of 91 °C for 4 s and 65 °C for 40 s during which the optical readings were taken on the CY5 channel. Experiments were run using an Applied BioSystems 7500 Fast Real Time machine (ABI 7500).(2)For the testing of biological samples (swab eluate) on the BioGene XF4 prototype instrument (XF4 prototype) the reaction volume was increased to 92 µL to maximize the sample input and hence sensitivity. The reaction consisted of 12 µL of nasal swab eluate, 45.2 µL of nuclease free water, 23 µL of 4* XF buffer, 3.22 µL of 10 µM forward primer mix (350 nm of each final conc), 3.22 µL of 10 µM reverse primer mix (350 nM of each final conc), 1.84 µL of 10 µM probe (200 nM final conc) and 1.84 µL of XF RTA enzyme. Additionally, these reactions varied from the smaller reactions by the addition of 0.45 µL of an RNA stabilizer (BioGene Ltd., UK) and 0.73 µL of 1 M NaCl, because the formulation was a research and development version of the buffer and ultimately these would be included in the 4* mix. The thermal cycling conditions were same as above.

#### 2.5.2. FMD Assay

RT-qPCR was carried out using the appropriate primers and probes, the BioGene XF reagent system in one of two assay formats. Unless stated otherwise, 25 µL reaction volumes were prepared as for the PPR assays. The assay conditions were as follows: 1 cycle of 93 °C for 5 s, 74 °C for 2 min, then 7 cycles of 93 °C for 4 s and 72 °C for 1 min, a single 93 °C for 10 s, and lastly, 35 cycles of 91 °C for 4 s and 65 °C for 40 s during which the optical readings were taken on the CY5 channel. The experiments were run using an ABI 7500 Fast Real Time instrument.

For the swab eluate testing using the XF4 prototype, the reaction volume was increased to the same 92 µL as the PPR assay and, the conditions were identical aside from a reduced probe concentration of 100 nM. The thermal cycling was performed identically to that of the ABI 7500 Fast Real Time instrument.

#### 2.5.3. Specificity Testing

To demonstrate the specificity of the approach, a series of 20 µL FMD reactions were set up according to a modification of the above protocol, 350 nM of each primer mix and 100 nM probe. These experiments were run on an ABI 7500 Fast. RNA extracted from tissue culture grown PPRV/Nigeria/75/1, swine vesicular disease virus (SVDV), vesicular stomatitis virus (VSV), Seneca valley virus (SVV) and respective positive and negative controls were used for the specificity test.

#### 2.5.4. Preparation of Swabs for Use in Prototype Pen-Side Assay

Nasal swabs were resuspended in 1000 µL nuclease-free water, vortexed briefly, before being centrifuged to collect the sample. From this, a 140 µL aliquot was taken for RNA extraction. The remainder was used directly in the newly designed assay or stored at −20 °C. 

### 2.6. RNA Extraction

Manual extractions were carried out using the QIAamp viral RNA mini kit (Qiagen) according to the manufacturer’s instructions. A total of 140 µL of viral suspension or nasal swab eluates were used for RNA extraction and, were eluted into a final volume of 60 µL. The RNA was either used immediately or stored at −70 °C until used.

### 2.7. “Gold Standard” RT-qPCR

The PPR real-time RT-qPCR was performed following the method described by Batten and colleagues [26] using the EXPRESS One-Step Superscript qRT-PCR kit (ThermoFisher Scientific). The FMDV real-time RT-qPCR was performed following the method described by Callahan and colleagues [27] using the EXPRESS One-Step Superscript qRT-PCR kit. Assays were carried out using an ABI 7500 Fast instrument.

### 2.8. Sample Matrix Effect Testing

Commercially available cotton buds were used for the collection of nasal secretions from the infected animals. The assay was performed as per the standard 25 µL reaction conditions for the two assays as above, with the exception that some of the water was replaced by nasal swab eluates at varying input percentages. For example, a 25 µL PPR reaction had the addition of 0, 8, 10, and 12% goat nasal swab eluate by volume, equivalent to 0 µL, 2 µL, 2.5 µL, and 3 µL in the 25 µL final volume. The sample used was 1 µL of unquantified genomic RNA extracted from the tissue culture-grown virus. For cattle nasal swab testing, an in house BioGene assay was used for detecting the NP sequence from Ebola virus, as the sample used Accuplex Ebola GP/NP reference (Seracare, Milford, CT, USA) had been quantified by digital droplet PCR. Twenty µL reactions were set up using 50 accuplex virions and the BioGene XF Ebola assay (BioGene Ltd., Kimbolton, UK) and run according to its protocol [30].

### 2.9. Viral Lysis Testing

These were performed as per the standard 25 µL reaction conditions for the two assays as above. The samples (viral suspension) were serially diluted ten-fold, and either 3 µL (PPRV) or 5 µL (FMDV) was used as template in the assay, determined by the initial titres of the respective viral cultures. For the reference tests, the extracted RNA was used using the protocols above and the calculated viral copy numbers added into the reactions were expected to be equivalent.

### 2.10. Assay Detection Limit Studies

For the newly developed assays 25 µL reactions were prepared in triplicate according to the protocols as above. The only variation was that the sample input for PPRV and FMDV were increased to 3 µL and 5 µL, respectively. 

The reference assays were set up as per the protocols above.

## 3. Results

### 3.1. Assay Optimisation

New assays were developed and optimized for the detection of PPRV and FMDV nucleic acid. The reference assays could not be adopted because (1) the existing primers had melting points that were too low for high temperature reverse transcription and (2) that genetic variation exists which is not accounted for by the reference designs which could lead to potential false negatives for some genotypes.

The closed tube process for direct detection consists of four stages: viral lysis, reverse transcription, cyclical reverse transcription and qPCR. The enzymes used in the BioGene XF reagent system are thermostable, necessitated as the viral lysis is performed using a combination of heat, solvent and detergent within the reaction itself. As a consequence, both reverse transcription and PCR will occur concurrently. In each denaturation cycle complementary DNA (cDNA) hybrids will denature and, the released RNA will be made available for further reverse transcription, the buffer being designed to stabilize the RNA. To take advantage of this, an additional intermediate step termed cyclical reverse transcription was added between the reverse transcription and qPCR stages, it is performed at a temperature and time where both reverse transcription and PCR can take place concurrently and thereby maximize the likelihood of turning low titre viral RNA into cDNA even in the presence of inhibitors. The initial experiments used a nominal concentration of primers and probes, 200 nM and 100 nM, respectively, and gradients of temperature were run for each of the final 3 steps to determine the optimal temperature ranges. Viral lysis was carried out at 91 °C for two minutes as this has been shown to work for all viruses that were previously tested and assists with thermal degradation of any inhibitors that may be present in the crude nasal swab eluate. An ABI 7500 Fast instrument was used for the assay development due to its ability to set thermal gradients and to process more number of samples concurrently than the four well XF4 prototype instrument.

The extant gold-standard reference assays were designed in regions of the viral genome that are highly conserved but also have considerable secondary structure. To overcome this, the XF reagent was used which can perform reverse transcription at a temperature up to 80 °C and to that end, the Tm of the FMD primers were 78 °C and those of PPR were 77 °C. Thermal gradient testing showed that the optimal reverse transcription temperature for the two assays was 74 °C and 72 °C, respectively. An example experiment for both the assays are shown in Appendix A, respectively, that shows very consistent RT activity across a wide range of temperatures. The new assays are completely complementary to over 99.9% of PPRV and FMDV sequences that were available in the GenBank database.

Similar testing was performed for the cyclical reverse transcription and qPCR steps (not shown). It was determined that at below 70 °C, no reverse transcription took place for FMD and, for PPR, it was 63 °C. In the case of the FMD assay, this could be due to the presence of a hairpin in the sequence located under part of the reverse priming site; RT taking place at a low temperature would likely encourage the hairpin to form before the primer could bind and drive reverse transcription. As the predicted Tm of both assays to DNA was around 70 °C, it was experimentally determined that the optimal temperature for the cyclical reverse transcription would be 72 °C, as both reverse transcription and PCR was taking place at the same time for both the assays. When the temperature of the cyclical reverse transcription step is close to that of the Tm of the primers, it is necessary to extend the time for this step to give sufficient time for efficient PCR to take place. Experiments investigating hold time were performed and it was determined that 90 s allowed the best compromise between PCR and reverse transcription and hence the earliest cycle threshold (Ct) values.

Lastly, the PCR itself was subjected to a gradient experiment and it was determined that 65 °C was a suitable PCR temperature for both the assays. The effect of primer and probe concentrations were studied as there were 13 individual primers in each assay and higher primer concentrations could potentially lead to non-specific amplification due to the presence of micromolar amounts of primer with high Tms. The optimal primer concentrations were determined by titration experiments and the final concentration (350 nM) of each primer was chosen as the best balance of early Ct but highest final observed relative fluorescence units (RFU). Experimentally both assays required 250 nM of equimolar primer mix to drive efficient RT in a 90 s hold as a minimum concentration. The optimal probe concentration experiments revealed that excess probe inhibited the FMD reactions potentially because of the very high Tm of the probe. The final chosen conditions for both the assays are outlined in the materials and methods section. 

### 3.2. Assay Specificity

The candidate assays were designed using VisualOMP, a package which uses a multi-state model to determine the likely primer and probe binding energetics [29]. VisualOMP has a functionality called ThermoBLAST which can screen newly designed primers and probes for their ability to generate potential off target amplification. Traditional BLAST analyses show sequence similarity but are unable to predict if these would impact assay performance. Essentially, ThermoBLAST takes the generated sequence hits and attempts to fold the assay primers and probes to these sequences to determine if any spurious amplicons could be generated. To confirm the accuracy of this approach, duplicate reactions were set up for the FMD assay, performed according to the routine protocol, and these were spiked with RNA extracted from FMDV, PPRV, SVDV, VSV and SVV grown in cell cultures and negative controls. Only the addition of FMDV RNA generated any signal (Appendix A), confirming the specificity of the assay design methodology.

### 3.3. Comparison of Assay Sensitivity with Gold-Standard RT-qPCR Test

Once optimal conditions were established, the relative sensitivity of the assays was established on the extracted viral RNA by comparison to the existing reference (RT-qPCR) tests. Ct values are generally not directly comparable between reagents, instruments, or assays and, the XF reagent is additionally able to skew Ct values by potentially turning a single RNA molecule into more than one cDNA copy (Appendix A). As a result of this potential pre-amplification, assays using the XF reagent are semi-quantitative detection assays that are intended to provide positive/negative diagnostic semi-quantitative results. The RNA extracted from viral suspensions was used from either the Sungri/96 (PPRV) or SAT 2 (FMDV) to establish whether the assays had sufficient sensitivity to be taken forward. The XF PPR assay was able to detect the same seven-fold serial dilution of the extracted Sungri/96 RNA as the reference assay (Appendix A), the purple curve, representing neat undiluted RNA, amplified too early in the XF reagent data to be able to baseline correctly. When the extracted RNA was used as the template, both assays were able to detect down to the same level of dilution and, therefore, it was decided that it was worthwhile to take forward the PPR assay to test the real archived animal samples.

For FMD assay, a similar procedure was followed where a serial dilution of extracted RNA (SAT 2 serotype) was used. The assays were set up and real-time PCR plots were generated from the both the reference and XF FMD assays. In this case, the sensitivity of the XF FMD assay appeared to be potentially higher than the reference assay with stronger amplification at the 10^−5^ dilution (purple curve in Appendix A). For both the XF assays, the cut-offs were over 25,000 RFU on the ABI systems, signal-to-noise of over 15:1, for future use on the XF portable instrument a slope and confidence score will also be used for automated results calling. Appendix A shows that the additional dilution to 10^−6^ nearly met the 25,000 RFU and signal-to-noise ratio requirements in all replicates, while no amplification occurred in the reference assay. The XF FMD assay was, therefore, taken forward for further testing using archived samples collected from infected animals.

### 3.4. Analytical Sensitivity

The approach used for the two XF assays was direct amplification from nasal swab eluate without prior RNA extraction. In order to perform the assays in the absence of nucleic acid extraction it was necessary to demonstrate that the virus was lysed and rendered amplifiable by this method. XF assays rely on a combination of heat, solvent and detergent to remove any lipid layers and break open the viral capsid while denaturing any proteins associated with the genetic material and finally protecting the released RNA from degradation. It was therefore necessary to demonstrate that the method could be applied to the direct detection of these two specific viral pathogens and that they were efficiently lysed and amplified.

For the PPR assay, viral cultures of four lineages (Ivory Coast, Nigeria, Sudan and Morocco) were added directly into the reactions in triplicate (Appendix A). Two different serial dilutions were tested, a 1:10 and a 1:100 dilution of the neat viral culture. In each case the virus was successfully lysed and amplified (Appendix A), indicating that the assay detected exemplar viruses from each lineage.

In the case of the FMD assay, viral cultures of six serotypes (A, Asia 1, O, SAT 1, SAT 2 and SAT 3) were used in 1:10 and 1:100 dilution. In each case the virus was successfully lysed and amplified (Appendix A).

The assay design was locked down and a final study was performed. Viral cultures of all exemplar viruses from four lineages of PPRV and six serotypes of FMDV were normalized to a concentration of 4.7 log_10_ TCID_50_/mL and ten-fold serial dilutions were prepared. The data is presented in Table 1 (PPRV) and Table 2 (FMDV), both assays were able to detect all four lineages of PPRV and six serotypes of FMDV, respectively. 

These data confirmed the findings that the PPR assay had similar sensitivity to the reference assay, with the benefit of not requiring prior nucleic acid extraction, and that the FMD assay outperformed the reference assay for serotypes A and Asia1 and detected all targets. The limit of detection for the designed PPR RT-qPCR across all lineages was determined to be between 0.5–0.7 log_10_ TCID_50_/mL which equated to a Ct value of 27.2–30.8, while the FMD RT-qPCR had a relatively lower limit of detection between 0.1–0.3 log_10_ TCID_50_/mL across all six serotypes with Ct value ranges of 30.67–34.51. Further study with larger number of biological samples using the assay in its final lyophilized format would be required for confirmation. The results from this study confirmed that the assays could detect exemplar viruses from all lineages/serotypes of PPRV/FMDV and, that the XF reagent rendered the viruses amplifiable.

### 3.5. Sample Matrix Effect Testing

Final testing was to be performed directly from nasal swab eluate and compared to the reference assays that use extracted RNA. The final target sensitivity was estimated to be below 5000 virions/mL as the literature suggested that viral quantity in a nasal swab might be expected to be at 10,000 virions/mL of eluate at the point at which the reference assays could be guaranteed to successfully detect the pathogens [28,31]. It was decided in the first instance that the elution to be made in water as opposed to a buffer which may have had its own impact on the assay. Testing was performed by resuspending the nasal swab in 1 mL of sterile water and determining the impact on the assay that could potentially arise from the presence of any inhibitors in the crude sample or arising from the cotton swab itself. In a larger scale study, it would be possible to investigate the impact of differing reaction and elution volumes; the goal here was simply to show that such a pen-side approach could have similar sensitivity to the lab-based reference method. Cotton swabs were chosen as these are freely available in resource limited countries and the archived animal study samples used cotton swabs. The assays had previously been validated to detect 50 copies of IVT RNA in the absence of inhibitors, the equivalent of 5000 virions/mL eluate if the swab is resuspended in 1 mL and 10 µL swab eluate is added into a reaction. Therefore, if the eluate could be shown to minimally inhibit the process, then proof-of-concept could be established, and testing can be taken forward to the archived samples.

Nasal swabs collected from goats on 0 dpc were used to make 1 ml of eluate and then spiked into an XF PPR assay at 0%, 8%, 10%, and 12%. No difference in the sensitivity was observed, indicating that the eluate could be tolerated at least up to 12% and, also that the cotton swabs were a suitable sampling medium in the reaction (Appendix A). For FMD, nasal swabs collected from cattle on 0 dpc were used to prepare the swab eluate. In this instance, an in-house control assay, detecting the Ebola NP gene (EBV XF assay, BioGene Ltd., Kimbolton, UK), was used as the target, a quantified standard that is used in human IVD diagnostics where the number of virions added was precisely characterized and at the proposed required lower limit of detection (LLOD) as similar reference material was not available for PPRV/FMDV. The Accuplex (SeraCare) Ebola GP/NP standard was added into the reaction at 50 virions per reaction and then spiked with up to 25% of the cattle nasal swab eluate (Appendix A). Minimal shifts (up to 0.2 Ct at higher inputs) were observed, however none were observed at up to 15% nasal swab input.

As a result, it was decided to carry out the final testing directly from the swab eluate as a 92 μL reaction containing 12 μL swab eluate as at this condition (13% by volume nasal swab) the tests were proven able to easily detect 50 copies of IVT RNA (corresponding to 4166 virions/mL) without interference by the biological matrix

### 3.6. FMD Assay Performance Evaluation

The final FMD assay performance evaluation compared the XF FMD assay direct from nasal swab eluate to the reference assay performed using the extracted RNA. The aim was to demonstrate that that the new XF assay had a similar performance to a lab test but without the requirement for prior RNA extraction. However, further studies would be required to ascertain the absolute sensitivity for this assay.

The reference FMD assay was carried out following the published protocol and the XF FMD assay was carried out as described in the materials and methods. For comparison it was estimated that the XF FMD assay would be adding approximately 10 virions more into its 92 µL reaction volume than the reference test when back calculating from the published reference method protocol and, assuming the virus was present at 10,000 virions/mL.

For FMD, the cut-off values to determine positivity were 40 cycles and 0.4 delta Rn for the reference assay run on the Applied Biosystems ABI 7500 Fast and 35 cycles and 500 RFU for the XF4 prototype instrument, reflecting that the protocol for the two assays are comparable between the two methods. The data showed that the XF FMD assay detected the presence of the virus in all animals at 1 dpc to 3 dpc whereas the reference assay detected positive in all animals at 2 and 3 dpc indicating that the XF assay is relatively more sensitive compared to the reference assay. On the XF prototype no data were acquired during the seven cyclical RT cycles and as such, six cycles were added onto the observed Ct value, since the first cycle would only have generated the second strand as opposed to performing PCR. The data were analyzed by averaging the fluorescence values corresponding to the CY5 probe then baselining and plotting on a graph of fluorescence intensity versus cycle number with a three-point rolling average applied to smooth the data and calling the Ct value during the second cycle where light increase was exponential.

The results of the study are presented in Table 3, which shows whether the target was detected at the indicated time point post challenge and the corresponding Ct values. The Ct values cannot be compared directly between the two methods and is only provided to give an indication of the viral load. 

The samples for the 12 challenged animals show that the XF FMD assay was able to detect the pathogen in all the samples at 2 dpc, in line with the reference assay. This demonstrated that the XF assay is equivalent to the reference assay, considering that all the samples were detected at the same time point that the reference assay had 100% analytical sensitivity.

There was good agreement generally between the two methods, with the XF assay having a possible tendency to detect the virus earlier in infection, evidenced by the fact that some 1 dpc samples were positive compared to the reference assay (Table 3). This was most likely attributed to the XF assay having a better limit of detection.

There were two examples of a sample that was detected by the reference assay and not by the XF assay; animals 12 and 8 were detected by reference assay at 3 dpc but not by the XF assay. There is a possibility that some nasal swabs may be inhibitory and in a field-testing situation multiple animals from the herd would need to be screened and so, this potential eventuality would be covered. These samples had, however, previously been detected by the XF FMD assay at 1 and 2 dpc, where this was not the case for the reference assay. Conversely, there were a number of time points where the virus was detected by the XF FMD assay and not by the reference assay, for example animals 6 and 10 at 3 dpc. At 1 dpc, there were a number of samples where the XF FMD assay appeared to be able to detect infected animals earlier than the reference assay, presumably resulting from the greater sensitivity of the assay. For the two unvaccinated animals, 1 and 2, the XF assay was positive at all time points. 

### 3.7. PPR Assay Performance Evaluation

The Ct values that were obtained in this study are presented in Table 4, which shows whether the target was detected at the indicated time point post challenge. When the two protocols were compared, the larger volume of the XF assay meant that approximately twice as many unprocessed virions would have gone into the final reaction as opposed to the RNA molecules into the 25 µL reference test, assuming 90% extraction efficiency. 

The results of the samples from the five challenged animals show that the XF PPR assay was able to detect the pathogen in all samples at 4 dpc (no sampling done on 3 dpc), in line with the reference assay indicating the performance of the XF assay is equivalent to that of the reference assay. There was good agreement generally between the two methods, with the XF assay having a possible tendency to detect the virus earlier in infection. This may be attributed to the higher sample input, potentially twice as many targets, and the XF assay having eight cycles in which RT could be performed and so potentially pre-amplifying low titred virus. In some of the later timepoint samples there was a trend towards later Ct values in the XF method than the reference assay. It cannot be discounted that some individual swabs have some slight inhibition which could skew the Ct but has not prevented target pathogen detection in the sample. Additionally, the instrument used to run the test was an earlier prototype that had simplified optics which may have lacked the sensitivity to detect small fluorescence changes over baseline and hence may have appeared to make Ct later than that observed on the laboratory instrumentation.

The only sample that was detected by the reference assay and missed by the novel XF PPR assay was animal 4276 at day 14, the possibility exists that this sample contained inhibitors that requires further testing using a larger number of samples.

At 2 dpc, the XF PPR assay detected more number of infected animals (4 out of five), but again this would need to be tested on a larger set of samples. If proven reproducible then it likely arises due to the doubled sample input and the ability to perform multiple rounds of reverse transcription, hence increasing the chance of amplifying when the input titre is very low.

## 4. Discussion

As countries move toward the global eradication of PPR by 2030 it is important that endemic countries have access to rapid and accurate diagnostic tests. Whilst countries who wish to retain their FMD-free status require sensitive testing during routine sero-surveillance to ensure potentially infected animals are not imported into the country. In this study, we designed primers for both PPRV and FMDV which targeted a conserved region of the viral genome with 100% complementarity to >99.9% of known sequences. This should allow all lineages/serotypes of the viruses to be detected, though the final direct evaluation was performed with a single lineage/serotype. We have developed two new RT-qPCR assays which achieved the same level of analytical sensitivity and specificity as previously published reference RT-qPCR assays; these assays did not require prior RNA extraction from the samples. Removing the RNA extraction step has multiple advantages, including (i) saving time between animal sampling and diagnosis, as well as (ii) eliminating the chance of cross-contamination during the extraction step, especially if multiple samples are being processed at the same time and (iii) lowers the chance of samples being mis-labelled during processing. As this was a proof-of-concept study, cell culture-grown virus and naïve swab spiking was used to ensure that the assay could perform to the accepted standard, as well as ensuring that the use of crude animal nasal samples did not inhibit the efficiency of the assay.

The XF assays demonstrated direct detection of viral nucleic acid from nasal swab eluates to a level that could have potential application in a field setting for diseases such as PPR and FMD. The XF reagent is also able to amplify directly from other crude sample types, including blood and urine, and so may have wider application in the detection of other animal pathogens [30]. An additional point for further study would be using the same assay direct from either oral or ocular swabs or blood as there is a viraemic stage to the disease, although nasal swab samples have been shown as the best sample type for PPR [28]. With further study, diagnostic panels could potentially be developed using the XF reagent that would enable in-field differential diagnostics, differentiating between diseases that present similar clinical symptoms. Additionally, it would be of benefit to study the relative sensitivity of smaller assay volumes, which would minimize the costs for use in poorer nations; the 92 µL reaction volumes used here exceeded the sensitivity required to match the reference assays. The commercial version of the instrument, the BioGene QuRapID XF1, will be fully portable and the assays will be deployed cold chain free, taking less than 40 min per test and so will be suitable for in-field and point of entry screening.

Isolates representing all four lineages of PPR (lineages I–IV) and the six serotypes of FMD (A, O, Asia 1, SAT 1–3) could be detected by their respective PPR or FMD assays. It should also be noted that the assays detected each lineage/serotype to the same level, displaying that the designed assay primers are not biased toward a single serotype/lineage or what is perceived to be the most “common” isolate of the virus; allowing the assay to be used anywhere where testing is required.

It was observed during the assay development that less starting material is required for the newly designed RT-qPCR compared to the currently used reference assays. The designed PPR assay used 2 µL of crude sample, which, during initial testing, was 2 µL of 4.7 log_10_ TCID_50_/mL virus, while the reference assay uses 3 µL of extracted RNA. During preparation, 100 µL of 4.7 log_10_ TCID_50_/mL virus was used for the extraction protocol, and the RNA was eluted into 60 µL water. This means 3.75 times more starting material was required for the reference assay in order to achieve the same level of sensitivity as the newly developed assay. For the FMD RT-qPCR, this increased to 12.5 times more RNA, as the assay specified 5 µL of extracted RNA compared to the 1 µL of sample/virus that is required to achieve the same sensitivity in the newly developed assay. Therefore, not only do the newly designed PPR and FMD assays achieve the same specificity and sensitivity as previously published “gold-standard” assay, but they are also capable of doing so with less sample input. 

Although this was a proof-of-concept study, we wanted to simulate real life sampling through the use of nasal swabs collected from PPRV/FMDV challenged animals. As observed for both FMDV and PPRV infected animals, all the animals were detected by two to four days post challenge and therefore, these assays would be able to detect the presence of the virus with confidence, before clinical symptoms manifest; making a clear case for its’ use during surveillance of both PPR and FMD.

In terms of suitability for the application, the research by Parida and colleagues [28] and Alexanderson and colleagues [31] shows an expected viral titre of 10,000 virions or copies/mL for nasal swab eluates by 4 dpc for PPR and 2 dpc for FMD. Clinical symptoms, however would not be expected to develop in all animals until day four for both diseases [31,32] and so these assays have the potential to detect infected animals that are not yet displaying clinical signs. The ultimate determination of the LLOD for the assays was outside the scope of this work as it would have required repeated trials to perform statistical analysis and establishment of titrated surrogates for the two viruses. However, both assays were validated down to 50 targets per reaction using IVT RNA, though the true LLOD would have been lower as the Ct values were 31 for PPRV and 30 for FMDV. As the sample input volume was 12 µL, this would give a putative LLOD of less than 3.6 log_10_ copies/mL nasal swab eluate and, therefore, below the initial design parameter of 10,000 virions/mL detection. This could be corroborated by the fact that in the final evaluation, all the samples were positive by 4 dpc and 1 dpc for the PPR and FMD assays, respectively, showing that the goal of at least matching the performance of the reference assays had been achieved.

The results presented here showed direct detection of the viral nucleic acid from nasal swab samples, which would allow detection and surveillance to be undertaken non-invasively. The advantage of non-invasive testing allows for veterinary assistants to sample animals without the requirement of licensed personnel, which could increase the number of animals that could be tested, as well as decreasing the cost of sampling per animal. If required, both assays can use whole blood as the starting material instead of oral/nasal swabs. We have observed during initial testing that blood has no inhibitory effect at low reaction percentage volumes (12–18%), the original development of the BioGene XF reagent having been designed for the direct detection of Ebola from whole unprocessed human blood [30]. The reagent has previously been shown to amplify from blood, plasma, serum, ocular and nasal swabs, viral transport medium, cerebrospinal fluid, saliva, and urine and, as such, should provide a suitable platform technology for the development of further veterinary diagnostic assays. 

The XF reagent contains an inhibitor-resistant thermostable enzyme mix that is capable of performing DNA- and RNA-directed DNA polymerization. The reagent contains a solvent and detergent mixture that, in conjunction with an initial high temperature hold, lyses the target virus and renders the genome amplifiable in the presence of the crude sample. An important feature is the ability of the mix to stabilize the released RNA and this, in combination with the thermostability of the enzymes, allows multiple rounds of reverse transcription to occur. This is termed cyclical reverse transcription and is shown schematically in S1. It is this ability that is expected to account for the sensitivity observed in this study. Well-designed assays using this approach are capable of detecting low numbers of unprocessed virions due to having multiple opportunities to generate an initial cDNA hybrid. This effect means that assays are semi-quantitative as high titres can show compressed Ct values as multiple cDNA may be generated from a single genomic target. This was observed in this study for example in Table 2 where the XF assay shows very early Ct at high viral titres in comparison to the reference assay. Another benefit of the approach is that regions such as 3D, the site of the FMD target assay, have strong secondary structure and in this study, we have shown reverse transcription at up to 78 °C is possible. High temperature reverse transcription can assist in amplification of these highly conserved regions with stable secondary structure and thus make the approach widely applicable for the detection of a wide range of viral pathogens.

In conclusion, we have developed novel closed-tube PPR and FMD pen-side assays with comparable levels of analytical sensitivity and specificity as the reference assays, without the requirement of prior RNA extraction. Both assays have the potential for rapid diagnosis of disease through the utilization of crude sample, which could have potential application to aid in both the PPR eradication program and continued surveillance of FMD. Future work will focus on testing the assays in-field and the creation of differential diagnostic panels that are capable of discriminating between diseases presenting similar clinical symptoms, such as PPR and FMD.

## Figures and Tables

**Figure 1 viruses-14-00835-f001:**
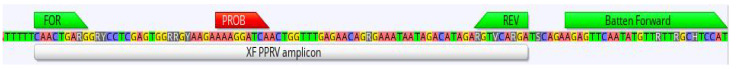
Location of the primers and probes of the newly developed PPR assay. The location of the forward primer of the PPR reference assay is shown to provide context. The alignment is to a consensus sequence of 300 nucleocapsid (N) gene sequences of PPRV aligned to accession number KY885100.

**Figure 2 viruses-14-00835-f002:**
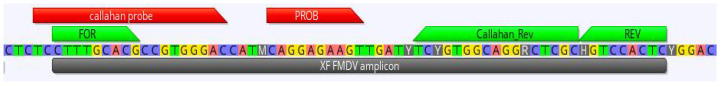
Location of the primers and probes of the newly developed FMD assay. The location of the primer/probe of the gold standard FMD reference assay is shown to provide context. The consensus is generated from the alignment of 1044 FMDV full genome sequences to accession number AF189157.

**Table 1 viruses-14-00835-t001:** Comparative Ct values that were obtained for PPR XF and reference assays. All four lineages of PPRV ((I—Ivory Coast/89, II—Nigeria/75/1, III—Sudan/72, and IV—Morocco/2008)) were normalized to a titre of 4.7 log_10_ TCID_50_/mL. Crude virus was used in the newly-designed assay (New), whilst RNA was extracted for use in the previously published reference assay (Ref) [26]. In both assays, the template was serially diluted neat to 10^−5^ to generate a standard curve. The samples denoted (-) refers to the sample failing to amplify.

	Ct Values							
	Lineage I		Lineage II		Lineage III		Lineage IV	
	New	Ref	New	Ref	New	Ref	New	Ref
1	17.45	18.81	21.68	20.64	19.37	25.1	22.1	24.43
10^−1^	20.319	22.5	22.4	23.73	21.2	28.2	24.12	27.53
10^−2^	25.82	25.74	25.82	27	23.04	31.3	25.82	30.41
10^−3^	26.8	29.4	26.8	30.5	25.9	34.73	26.8	33.84
10^−4^	30.87	32.68	30.8	33.9	27.2	37.49	30.87	39.56
10^−5^	-	-	-	-	-	-	-	-

**Table 2 viruses-14-00835-t002:** Comparative Ct values that were obtained for FMD XF and reference assays. A total of six serotypes of FMD viruses ((A (A22/Iraq24/64), O (O1/Manisa), Asia1 (Asia1 Shamir), SAT1 (SAT1/TAN/22/2012), SAT2 (SAT2/Eritrea/98) and SAT3 (SAT3/ZIM/83)) were normalized to a titre of 4.7 log_10_ TCID_50_/mL and the unprocessed virus was used in the newly designed XF assay (New) or alternately, RNA was extracted and serially diluted 1–10^−7^ that were used in the previously published reference (Ref) assay [27]. Ct values for each assay type are compared to each other based on serotype. The samples denoted with (-) are samples that failed to amplify. Standard curves follow the hypothesized linear relationship between Ct value and dilution.

	Ct Values											
	A		O		Asia1		SAT 1		SAT 2		SAT 3	
	New	Ref	New	Ref	New	Ref	New	Ref	New	Ref	New	Ref
1	12.26	16.41	13.98	13.95	14.19	19.46	14.45	15.57	15.41	16.63	13.27	12.8
10^−1^	16.45	19.54	17.82	17.99	18.19	23.77	18.97	18.59	20.27	19.82	16.9	15.7
10^−2^	19.76	23.8	20.99	21.19	22.94	-	21.95	21.68	23.46	22.76	20	N/A
10^−3^	23.05	28.66	23.72	25.55	26.46	-	24.67	24.56	26.97	25.21	22.96	21.94
10^−4^	23.98	-	25.27	27.92	29.51	-	27.25	27.63	30.1	27.93	24.04	25.26
10^−5^	25.289	-	28.04	32.3	32.37	-	28.41	31.01	31.18	29.7	26.86	27.55
10^−6^	29.07	-	30.88	32.95	-	-	30.13	32.88	29.17	32.9	29.1	29.05
10^−7^	30.67	-	34.51	28.27	-	-	33.98	32.41	32.07	30.16	31.58	30.53

**Table 3 viruses-14-00835-t003:** The Ct values obtained for the XF FMD assay and the reference assay for the time course experiment for the 12 infected animals at 0 dpc to 3 dpc. No Ct indicates that no amplification occurred for a given time point.

XF FMD Assay Ct Values												
Animal No.	1	2	3	4	5	6	7	8	9	10	11	12
0 dpc	No Ct	No Ct	No Ct	No Ct	No Ct	No Ct	No Ct	No Ct	No Ct	No sample	No Ct	No Ct
1 dpc	32	33	33	33	33	32	33	30	33	32	34	30
2 dpc	28	27	30	31	31	32	30	33	29	31	33	29
3 dpc	26	26	29	30	30	32	No sample	No Ct	31	30	31	No Ct
**Reference Assay Ct Values**												
**Animal No.**	**1**	**2**	**3**	**4**	**5**	**6**	**7**	**8**	**9**	**10**	**11**	**12**
0 dpc	No Ct	No Ct	No Ct	No Ct	No Ct	No Ct	No Ct	No Ct	No Ct	No sample	No Ct	No Ct
1 dpc	29.6	39	39	No Ct	No Ct	30	No Ct	No Ct	34	27.4	No Ct	27.4
2 dpc	27.5	21.5	39	24.6	35	39	29.5	No Ct	23.5	25.3	31.8	No Ct
3 dpc	No Ct	21.4	28.8	27.2	39	No Ct	No sample	32.8	28	No Ct	29.7	31.8

**Table 4 viruses-14-00835-t004:** Ct values that were obtained from the time course experiment for the five infected animals from 0 dpc to 14 dpc for the XF PPR assay and the reference assay. No Ct indicates that no amplification occurred for some of the time points.

	XF PPR Assay Ct Values				
Animal No.	197	7713	4276	4231	4235
0 dpc	No Ct	No Ct	No Ct	No Ct	No Ct
2 dpc	32	29	No Ct	30	32
4 dpc	29	32	32	30	30
5 dpc	26	30	33	28	25
6 dpc	25	27	31	26	25
7 dpc	23	26	30	no sample	24
8 dpc	20	23	27	no sample	24
9 dpc	no sample	25	27	no sample	25
10 dpc	24	24	25	no sample	27
12 dpc	28	27	28	no sample	29
14 dpc	no sample	27	No Ct	no sample	30
	**Reference PPR Assay Ct Values**				
Animal No.	197	7713	4276	4231	4235
0 dpc	No Ct	No Ct	No Ct	No Ct	No Ct
2 dpc	No Ct	31.5	No Ct	No Ct	No Ct
4 dpc	26	33.8	34	27.9	27.4
5 dpc	26.9	29.3	32.7	26	23
6 dpc	24.7	29.5	29.6	24	21.7
7 dpc	20.2	25.7	30	no sample	23.3
8 dpc	19.9	21.3	24.8	no sample	20.8
9 dpc	no sample	21	21.3	no sample	22.5
10 dpc	24	22.1	24	no sample	24.7
12 dpc	25.2	25.4	26	no sample	25.3
14 dpc	no sample	25	24.9	no sample	25.4

## Data Availability

Not applicable.

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
