# Peer review of "Development and Evaluation of Molecular Pen-Side Assays without Prior RNA Extraction for Peste des Petits Ruminants (PPR) and Foot and Mouth Disease (FMD)"

_viruses, 2022, doi:10.3390/v14040835_

Round 1

Reviewer 1 Report

The manuscript entitled “Development and evaluation of molecular pen-side assays without RNA extraction for Peste des petits ruminants (PPR) and Foot and mouth disease (FMD)” describes the design and the preliminary evaluation of novel potential field tests for the detection of PPRV and FMDV based on realtime RT-PCRs, targeted on the same genomic regions as the respective reference laboratory assays, but using different probes and combinations of multiple primers. The availability of on-site diagnostic systems for these two diseases is a very topical goal and their exploitation is useful in numerous situations, therefore studies addressed at increasing their availability and/or improving their performances are worthy of publication.

However, in the current format the manuscript has some critical issues that should be addressed to make it suitable for publication.

MAJOR REVISION

The text is exaggeratedly long and this raises the risk of dispersing the reader in excessive details, especially in the preliminary phases of tests optimization, leading to miss the attention on essential results. There are several repetitions or redundancies, both in Materials&Methods and Results chapters, that I think can be avoided, thus making the text more concise and easier to follow.

  • For example, sub-chapters 2.5.1 (PPRV assay) and 2.5.2 (FMDV assay) could be condensed in a unique sub-chapter, since lines 206-209 and 215-222 for the PPRV assay are respectively identical to lines 230-234 and 240-247 for the FMDV assay. Furthermore, also the text describing assays conditions is repeated twice in each sub-chapter (PPRV assay conditions in lines 209-214 are identical to lines 223-227; FMDV assay conditions in lines 234-239 are identical to lines 248-252). I suggest to merge these sub-chapters, describing the common conditions of the assays once and highlighting where differences exist instead of repeating all the procedures twice.
  • The paragraph of “Specificity testing” (lines 253-263), only relevant to the FMDV assay, repeats once again the same protocol described just above, therefore I think the paragraph is redundant and can be deleted; contextually, the sentence at lines 381-382 in chapter Results can be completed with the unique missing information, just added the underlined words in the following example: “To confirm the accuracy of this approach, duplicate reactions were set up for the FMDV assay, performed according to the routine protocol, and these were spiked with RNA extracted from FMDV, PPRV, SVDV, VSV, SVV grown in cell cultures and negative controls”.
  • Similarly, sub-chapters 2.9 and 2.10 do not provide specific additional information, the protocols applied are those already described in previous sub-chapters, with the only variations of the template volume balanced with water; these minor variations can be specified when reporting results, with relevant reasons, while deleting sub-chapters 2.9 and 2.10 in Materials & Methods.

Sub-chapter 2.2: here some more details of the experimental infections is needed, with indication of the lineage/serotype used for animals infection and for FMDV the schedule of vaccination and challenge. Occurrence time of clinical symptoms should also be indicated.

“Inhibition testing” is an unusual way to describe studies aimed at investigating the possible adverse effect of the sampling materials, and it is not readily understood; “Evaluation of matrix effect” is more appropriate.

“Routine testing” etc (sub-chapter 2.5): the terminology used to define the tests along the manuscript (routine test, direct test, reference assays, gold standard, etc …) is sometimes not clear and make it difficult to catch which conditions have been used/applied. For ex. for experiments of specificity (2.5.2), the test applied is the so-called “routine test” but with the variation of the reaction volume (20 µL, as specified at line 254, instead of 25 µL as per line 230), is it correct?

The denomination “routine testing” can generate misinterpretation, since the thought goes to tests used routinely for diagnostic scopes; maybe “test optimization studies” or a similar concept is more appropriate.

For sake of clarity, would not it be possible to re-name also the “testing for nasal swabs” (2.5 and further) as “prototype-penside assay” used to analyse nasal swabs, or something similar?

Additionally, the novel assay is sometimes named as “direct assay or direct method” when compared with the reference gold-standard one, sometimes as BG XF assay or BG assay (lines 590-591) or BGR-BGR XF assay in tables 1 and 2. I suggest to identify it with a unique, unequivocal denomination.

For a better comprehension it should also be explained why all experiments for assays optimization were run as “routine testing” in Applied BioSystem 7500 machine instead of directly with the BioGene XF4 instrument.

It is not clear to me if the design of new primers and probes, albeit in the same region as the gold-standard tests, and the use of multiple primers was necessary to adapt the new tests to the pen-side format or if there are other reasons. Would it have been possible to use only the same primers / probes of the gold-standard tests which are extensively validated and globally used? Can the authors explain the choice?

Unfortunately, the quality and consequently the legibility of the figures is insufficient. I have not been able to read captions and legends in most of the figures/graphs. It is unusual for scientific papers to directly report the images returned by an instrument, instead elaborated/processed results are shown. In my opinion, most of figures are redundant; one gets the impression that all the experiments performed in the optimization phases have been reported in detail as in a workbook rather than selecting the essential data for a scientific publication. It is advisable to condense results by selecting the most important data. For example, I believe that figures 3 to 10 are not necessary, if the authors really care they can propose them as supplementary materials, provided that the quality / readability is improved (in some of them it is almost impossible to distinguish too many different curves/conditions in the same plot using colors that appear very similar), but I think that reporting essential results only in the text is more adequate.

Sub-chapter 3.3. I think that rather than “determination of assay sensitivity” the experiment described is a “comparison of assay sensitivity with gold-standard tests”; in fact, dilutions of RNAs extracted from viral suspensions are tested and results of the novel and gold-standard assays are compared, but the amount of detected RNA is not known/quantified.

Sub-chapter 3.4. This sub-chapter can be simplified. As far as I understand, the experiments described in figure 8 and 9 are also more extensively repeated in experiments whose results are reported in the supplementary tables S1 and S2. Indeed, results in Table S1 and S2 demonstrate either that virus lysis occurs without RNA extraction, and above all provide data on analytical sensitivity of the assays, as viral suspensions with known infectious titers were analyzed in a wider range of dilutions. Therefore, experiments reported in figures 8 and 9 are redundant and can be eliminated, while more emphasis to results reported in Table S1 and S2 should be given, moving these tables in the main manuscript, and not as Supplementary material.

However, though not indicated in Materials & Methods and Results, the tables legend states that for the reference (gold-standard) assays RNA was extracted from the net culture suspensions and then serially diluted, whilst I think that the correct approach for the comparison with the new XF assays would have been to extract RNA from each of the same serial dilutions of the virus suspensions which were directly examined in the XF assays. I propose that authors repeat this experiment which I think is one of the most critical and informative of the comparative study.

This 3.4 sub-chapter could be better titled “Analytical Sensitivity”.

Sub-chapter 3.5. The use of another test detecting Ebola NP gene in place of the FMDV assay to evaluate potential interference by the biological samples is not convincing for me. Can the authors use the target test or explain why the FMDV test was not used?

Sub-chapters 3.6 and 3.7. Here, too, the text can be shortened since there are several repetitions and the results can be described more concisely. The two subchapters can be combined into a single one, so that description and comments can be unified and not repeated. For ex. the concepts expressed at lines 521-523, 606-608, 619-623 are valid for both assays. The sentence covering lines 553-556 is not necessary as the information is given with more details below. The concept between lines 568-570 has already been expressed above (lines 558-560). The conclusion at lines 573-578 is more proper for the discussion, where it is actually already reported, and anyhow it is also repeated below for the PPRV assay at lines 624-626. The text at lines 586-589 and 595-600 in sub-chapter 3.7 is also present in the previous sub-chapter. For the above reasons I propose to condense the two sub-chapters in a single one.

FURTHER MINOR COMMENTS/REVISION

Line 71: please delete “across much of Africa, the Middle East and Asia. South America recently controlled FMD by vaccination”, because the information is also given below.

Line 75, Introduction: FMDV infection persistence for up to 3.5 years was only seen in buffalos and is a rare event, please modify.

Line 84: The statement “Serotype C is confined to the Indian sub-continent but has not been identified in animal samples since 2006” is incorrect. Serotype C caused outbreaks also in Europe, Africa and South America but is now considered extinct, with last detection in Kenya and Brazil in 2004 (there is a recent review published in 2021 by Virus Evolution).

Line 88: “This outbreak cost many European countries their FMD-free status”: please change many with some, because in addition to UK only The Netherlands and France experienced outbreaks in 2001.

Line 90: should be “FMD surveillance”, not FMD sero-surveillance.

Lines 103-105: at the end of Introduction it would be useful to introduce the new assays shortly describing the principle of the novelty and why/how they can be exploited as on-side test.

Sub-chapter 2.1: “The Vero DogSLAMtag (VDS) cells were used for growing PPRV” and “BHK21 cells were used for growing FMDV” can be deleted, since repeated also below in the same sub-chapter.

Sub-chapters 2.3.1 and 2.3.2: In fig 1 and 2 the exact location of the forward and reverse primers for the new assays does not seem to match with positions indicated in the text (395-477 for PPRV and 6823-6884 for FMDV) and this may lead the reader into confusion.

Sub-chapter 2.4 Assay optimization: it would be useful to specify the viral strains (genotype/serotypes) corresponding to the accession numbers used to obtain IVT RNAs.

The definition of “Batten assay” and “Callahan assay” used along the text and in figures legend is not so appropriate particularly for readers not very familiar with these assays; I would suggest to refer to them as “gold-standard assay described by … et al.”, or “assay described by … et al. and adopted as gold standard”, or “assay described by … et al. and selected as gold standard”.

The instrument used for the reference realtime RT-PCR is named in different ways, Applied BioSystem 7500 Fast Real Time PCR machine - Applied BioSystem 7500 - ABI 7500 - 7500 instrument. Please use a uniform denomination.

The sentence at lines 393-395 is not well expressed and can generate doubts: it is not clear to which test is referred the affirmation: “these are intended to be … “.  It should be referred to XF reagent test (“designed as semi quantitative detection assay”), but in that position it seems to be referred to the more standard qPCR?

Line 397, and legend for fig 6 and 7: useful to specify “RNA extracted from viral suspensions”

Legend table 1 and 2 (Supp): 4.7 TCID50 should be 4.7 TCID50/ml

Line 474 and 475:  TCID50 ml -1 should be corrected in TCID50/ml

Sentence at lines 510-512. The second part of the sentence does not sound clear to me; could it be: “As a result, it was decided to run the final direct from eluate testing as a 92μl reaction containing 12μl swab eluate as at this condition (13% by volume nasal swab) the tests were proven able to easily detect 50 copies of IVT RNA (corresponding to 5,000 virions/ml) without interference by the biological matrix”?

Line 537. “where the reference assay was only positive at 2 and 3 dpc” does not correspond to data in the table, that indicates positivity at 1 and 2 dpc. The abbreviation dpc is used without previous indication of the full denomination.

Line 561-562: please check, there are two animals not a single one, 4990 in addition to 7162, which are detected at 3 dpc by the reference test and not by the XF assay. 

Lines 565 and 571: low titre or higher titre referred to the output of these reactions is not a proper definition, as the Ct values do not represent “titres”.

Line 633: The first sentence should be mitigated as very few samples have been examined in parallel, or should be limited to “analytical” specificity and sensitivity.

Lines 633-638: the specific application indicated here for the FMDV novel assay is not proper.

Line 640, “allowed all serotypes/lineages of the viruses to be detected”: actually only a single strain representative for each lineage/serotype has been evaluated, this should be specified.

Line 655:  the sentence “With further study diagnostic panels could potentially be developed that would enable in-field differential diagnostics”. I don’t understand the context.

Line 677: see comment above for line 640.

Line 671 and 673: TCID50 4.7 ml−1, shouldn’t it be corrected as TCID50 log10 4.7/ml?

Line 692-694: rather than “a larger number of samples” it is a number of repeated trials that would be necessary to confirm the data, as usually what is needed to study the LLOD (in other words the analytical sensitivity) is a titrated sample, with known analyte concentration.

Line 698: “sub 10,000 virions/ml”, why “sub”?

Line 699: all samples came back as positive by 3dpc and 2dpc for the PPR and FMD assays 699 respectively. Should not it be “by 4 dpc and 1 dpc” according to results in the tables?

Line 704: health workers is perhaps a mediated definition that is more suited to the human sector.

The Supplementary Figure 3 is only mentioned in the discussion (Line 722); authors could refer to it also at the end of introduction, when presenting the novel assay principle, and again along the manuscript, when recalling its capability to perform RT and PCR reactions simultaneously.

Line 723: is there a reference to support the statement “capable of detecting 5 actual virions or less”?

Line 727: is it correct the reference to Table 1 here?

Line 735: again, the statement should be limited to analytical sensitivity and specificity. See comment above.

Line 738-740: see comment above, the statement does not seem in a proper context.

Legend for fig.11: I guess that 11(D) corresponds to 2 dpc, information is missing in the figure legend

Legend for tables 1(S) and 2(S): please substitute the provisional memo in legends for Table 1 and 2 with the identification of the virus strains actually used and report them also in the relevant paragraph 2.1 of Materials and Methods

Finally, there are some typo mistakes, some oversights, and improperly used punctuation along the manuscript. Authors are invited to carefully check it.

Author Response

Uploaded

Reviewer 2 Report

The manuscript by Edge et al., entitled "Development and evaluation of molecular pen-side assays without RNA extraction for PPR and FMD" described two new in-field assays for confirmatory diagnosis of two OIE notifiable diseases of livestock. The manuscript described the optimisation, analytical sensitivity and specificity and limited of detection of the assays. The novelty of the assays is the lysis buffer that allows reverse transcription and qPCR directly from the samples. 

The manuscript is well described and the topic was thoroughly and chronologically investigated.

Author Response

uploaded

Round 2

Reviewer 1 Report

The suggestion to present the results in a more concise and essential text by selecting those data which are more relevant, without redundancies and excessive details, was not followed, the structure and the length of the revised manuscript has remained unmodified; however, I acknowledge that the authors have recognized and solved several other highlighted criticisms.

Nevertheless, a revision to correct some inaccuracies and improve wording in some sentences is still necessary. Therefore, I invite authors to check more carefully the whole text and relevant figures; some (but not all) examples of needed revision are suggested below.

  • Punctuation and typing errors: various oversights remain, here are some (but not all) examples: line 46: a dot is opportune after the ref [1,2,3]; line 49: insert comma after PPR, line 59: delete the double comma after PPRV, line 128: at the beginning of the sentence, after a period, numbers should be written in full text, line 234 capital letter after the period, line 235: spacing missing between 25µl and reaction, µl should be µL along the manuscript, wherever TCID50/ml is used the number 50 should be subscript, in line 435, 1:105 the number 5 is exponent and must be superscript, other similar inaccuracies are at lines 390, 394, etc…. Please also check how the temperature degree symbol is indicated along the manuscript and in supplementary fig.1, sometimes it appears as a zero (for ex. in lines 217-218, where it appears as 930C and 720C, etc, see lines 236-238, see figS1 indicated with the letter C alone, …).
  • Line 58: not sure of the sentence grammatical construction, in particular for the position of “which has …. Could it be? “The disease has a high level of morbidity and mortality and transmission occurs through close contact of infected and naïve animals.”
  • Line 70: “across much of Africa, the Middle East and Asia” can be deleted as repeated at the following lines 75-76, where the reference n. 12 can be moved.
  • Line 117 and 119: “for each lineage of PPR”, “for each serotype” (add V to PPR and insert FMDV before serotype).
  • Line 133-134: the information present in the previous version was lost, useful to reinsert the underlined text “… nasal swabs were collected from experimental goats …”
  • Line 138: delete “N-gene” because it is in contrast with that sentence and delete “(n = 300)” because this info is provided below at line 141.
  • Line 142-143: please delete “with particular focus on the N gene region”, as it is redundant.
  • Line 222: please quote the source of the XF4 prototype by specifying “… on the BioGene XF4 prototype instrument”. Subsequently you can use an abbreviated form indicating it in brackets the first time, like XF4 prototype, if preferable.
  • Line 241: please add “using the XF4 prototype” after “For the swab eluate testing …”
  • Applied BioSystems 7500 Fast Real Time machine (ThermoFisher Scientific): please check again, it appears first at line 220, followed by line 239, but in lines 243, 248, 311, 391 and others it is not indicated in the same uniform way but with various definitions. I suggest to use the full name with an abbreviation in brackets the first time and then use the abbreviation only; not needed to repeat the source each time (alternatively given as ThermoFisher (line 239) or Applied Biosystems US (line 248)).
  • Lines 317 and 320: please check singular and plural use (for ex. in these sentences: the Tm of the FMDV primers was 78oC and those of PPRV were 77oC, An example experiment are shown).
  • A Title for table 1 and table 2 is missing
  • I think that the paragraph added into sub-chapter 3.1 to explain the reasoning for using combinations of new and multiple primers was inserted in an inappropriate position, better to move it at the beginning of 3.1, and I propose to use the sentences/wording used in the response to reviewer which are clearer and better written than the inserted paragraph.
  • I suggest to leave the paragraph 3.2 as in its original version: since the authors decided to maintain the paragraph of assay specificity also in materials and methods, there was no need to repeat information also in the chapter of Results.
  • Line 378: was the PPRV Morocco 2008 strain used in this and all experimental trials or was the Sangri/96 used, as indicated in the subsequent line 381 and in the relevant fig 5? The same discrepancy is present between the text in paragraph 3.4 and fig.S7.
  • Line 411: results shown in fig S7 support that there is a linear relationship between target dilution and assay signal but not that the assay was equally sensitive for the four lineages; there is in contrast an evident difference between amplification curves, at least for lineage IV represented by Sangri strain (or was the Morocco strain, please check?). Since the viruses concentrations were not normalized in this experiment, the observed difference could be caused by an initial difference in target concentrations, so I believe that in these conditions it is not correct to conclude that there is evidence of equal sensitivity for each represented lineage.
  • Line 427-428: check the sentence: “would be required” is repeated twice.
  • Table 1 and 2: “BGR assay” appear the first time in the columns header of these tables, please explain the acronym and uniform columns header “BGR assay” or “BGR XF assay”; Log10 is missing in legends when indicating the virus titre and please pay attention to superscript and subscript, and to singular plural use at lines 435-436.
  • Regarding the authors’ answer to my observation on the methodology adopted to compare analytical sensitivity of the new assays with that of the gold-standards (sub-chapter 3.4), I think there has been a misunderstanding as the answer seems out of context to me. Of course it is well clear that the XF assay is not designed to amplify from extracted RNA! But in a direct comparison of the analytical sensitivity of two assays the common and most appropriate approach is to apply the two methodologies to the same quantified samples, in this case the same serial dilutions of the crude viral suspensions. Anyhow, I am not going to argue on the chosen approach further.
  • Line 456: please check the new insertion of the adverb “potentially” in the sentence, perhaps it should be “potentially present”?
  • Line 466: the sentence “Goat nasal swabs collected at 0 days post challenge (dpc) before virus or vaccine has been administered used …” should be corrected as “Goat nasal swabs collected before virus or vaccine has been administered were used …” : the term challenge for vaccine administration is not appropriate, and “were” was missing. The extended form of days post challenge with its abbreviation can be move at line 471.
  • Line 474: please indicate the extended name of LLOD before using the abbreviation.
  • Legend of fig S9: please specify in the legend what is the test B and that its target was 50 copies of IVT RNA; also clarify that the addition of 50 targets per reaction is only referred to test B, whilst (according to the specific paragraph in Materials and Methods) the sample used for the PPRV reaction was 1μl of unquantified genomic RNA extracted from the tissue culture grown virus.
  • The sub-chapters 3.6 and 3.7 remained almost unmodified and still very long. For example, in contrast with authors’ answer the sentences at the current lines 520-523, 535-538, 586-589 were not removed and text adapted. In Fig.3 all labels and axes titles are unreadable, is this figure really necessary, given that the full set of results for all animals is more clearly presented in table 3?
  • In contrast to the response of authors (who declare “the sentence has been mitigated to analytical sensitivity and this is now used throughout the document”), the first sentence in the discussion was not changed (the referee had suggested to mitigate the statement since very few samples have been examined in parallel, or alternatively to limit the statement to “analytical” specificity and sensitivity).
  • Similarly, in contrast to the response of authors (who answered “This line has been omitted”) the unproper application indicated at the current lines 588-590 for the FMDV novel assay remained unaltered.
  • Line 679: still quoted fig.S3, now S1?
  • Line 684: please check again the reference to Supplementary table 3, shouldn’t it be table 1 or 2? There are not Supplementary tables.
  • The definition of most figures, though moved to supplementary materials, remained poor with labels and titles of the axes unreadable.

Author Response

Dear Reviewer,

Thank you very much for your time and deep review. We met all the points you raised and we appreciate your review. Please find the rebuttals where we have described how we met all the points.

Kind regards
